# NLRP3 Inflammasome Is Involved in Cocaine-Mediated Potentiation on Behavioral Changes in CX3CR1-Deficient Mice

**DOI:** 10.3390/jpm11100963

**Published:** 2021-09-27

**Authors:** Ming-Lei Guo, Ernest T. Chivero, Shannon E. Callen, Shilpa Buch

**Affiliations:** 1Department of Pathology and Anatomy, Eastern Virginia Medical School, Norfolk, VA 23507, USA; 2Department of Pharmacology and Experimental Neuroscience, University of Nebraska Medical Center, Omaha, NE 68198, USA; ernest.chivero@unmc.edu (E.T.C.); scallen@unmc.edu (S.E.C.)

**Keywords:** cocaine, microglia, neuroinflammation, CX3CR1, NLRP3 inflammasome

## Abstract

Microglia, the primary immunocompetent cells of the brain, are suggested to play a role in the development of drug addiction. Previous studies have identified the microglia-derived pro-inflammatory factor IL1β can promote the progression of cocaine addiction. Additionally, the activation status of microglia and “two-hit hypothesis” have been proposed in the field of drug addiction to explain how early life stress (ELS) could significantly increase the incidence of drug addiction in later life. However, the mechanisms underlying microglia prime and full activation and their roles in drug addiction remain greatly unexplored. Here, we employed CX3CR1-GFP mice (CX3CR1 functional deficiency, CX3CR1−/−) to explore whether primed microglia could potentiate cocaine-mediated behavioral changes and the possible underlying mechanisms. CX3CR1−/− mice revealed higher hyperlocomotion activity and conditional place preference than wild-type (WT) mice did under cocaine administration. In parallel, CX3CR1−/− mice showed higher activity of NLR family pyrin domain-containing 3 (NLRP3) inflammasome than WT mice. Interestingly, CX3CR1 deficiency itself could prime NLRP3 signaling by increasing the expression of NLPR3 and affect lysosome biogenesis under basal conditions. Taken together, our findings demonstrated that the functional status of microglia could have an impact on cocaine-mediated reward effects, and NLRP3 inflammasome activity was associated with this phenomenon. This study was consistent with the two-hit hypothesis and provided solid evidence to support the involvement of microglia in drug addiction. Targeting the NLRP3 inflammasome may represent a novel therapeutic approach for ameliorating or blocking the development of drug addiction.

## 1. Introduction

Drug addiction has been well acknowledged as a neuroplasticity disorder. In the last three decades, most investigations have been focusing on the changes on intra- and inter- neuronal signaling induced by abused drugs to explain their addictive properties. However, till now, no FDA-approved treatment, to our knowledge, is available to reverse/block drug addiction, especially for cocaine addiction. This dilemma indicated that, in addition to neurons, other types of brain cells, such as glia, could also play critical roles in drug addiction through glia–neuron crosstalk. Indeed, psychostimulants including cocaine activate microglia in vitro and in vivo [1,2]. Increased microglial activation has been consistently found in various stages of cocaine addiction [3,4]. Microarray analyses from mouse brains have implicated cocaine- or methamphetamine-mediated upregulation of the expression of pro-inflammatory factors such as IL1β, IL6, TNFα, and CCL2 in reward-related regions, such as the prefrontal cortex and nucleus accumbens [5,6]. Increased IL1β expression in the ventral tegmental area was critical for cocaine-mediated behavioral changes including conditional place preference (CPP) and self-administration [7]. In addition, minocycline, an inhibitor of microglial activation, is capable of blocking reward-related behavioral changes induced by abused drugs [8,9]. Impaired microglial activation due to Toll-like receptor 3 and 4 deficiency blunted cocaine-mediated behavioral changes [10,11]. Furthermore, cocaine self-administered monkeys showed a strong inflammatory response in the NAc [12]. The brains of addicts have also revealed a close association between neuroinflammation and stimulant addiction with a significant increase in the number of activated microglia [13,14]. Taken together, all these findings underscore the critical roles of glial and neuroimmune mechanisms in drug use and abuse [15].

Microglia are the residential macrophages in the brain and play crucial roles in determining neuroinflammation levels. At basal conditions, microglia reveal highly ramified and long processes to sense and survey the local environment. Once activated, these cells could quickly change their morphology with thickening and shortening processes, decreased branch numbers, increased soma volume, etc. Along with such striking morphological alterations, microglia increase the production and release of a plethora of cytokines and chemokines including Il6, TNFα, IL1β and CCL2, resulting in elevated neuroinflammation [16]. Such immune-related molecules are capable of stimulating the neighboring neurons and enhancing neuronal excitability which underlie the crosstalk between microglia and neurons.

NLR family pyrin domain-containing 3 (NLRP3) belongs to the superfamily of pattern recognition receptors [17] and is abundantly expressed in microglia compared to other CNS cells. Upon stimulation, NLRP3 recruits the apoptosis-associated speck-like protein containing a CARD (ASC) to form the inflammasome, resulting in the increased production of mature IL1β and IL18 [18,19]. IL1β could initiate multiple inflammatory pathways and drive inflammatory responses, which ultimately, modulates neuronal excitability and reward-related behavioral changes [20]. Abnormal NLRP3 inflammasome activity is indeed involved in multiple neuroinflammatory diseases, including Alzheimer’s disease (AD) [21], multiple sclerosis (MS) [22], as well as in brain injury [23] and viral infections [24]. The unique feature of NLRP3 signaling is that the full activation of this pathway needs two signals: signal 1 increases the transcription of IL1β (prime) and signal 2 leads to the formation of NLRP3 inflammasome, resulting in the increased production of mature IL1β production [17].

In the adult brain, CX3CR1 is predominantly expressed in microglia, restraining them in quiescence [25,26]. Previous studies suggested that NLRP3 signaling was negatively regulated by CX3CR1 [27]. CX3CR1 deficiency could prime microglia, resulting in stronger immune response on the following stimulus. As evidence, CX3CR1-GFP mice where CX3CR1 was replaced by the insertion of GFP showed increased LPS-induced neuroinflammation, neurotoxicity, and behavioral changes [28,29]. So, CX3CR1-GFP (CX3CR1−/−) mice could represent a suitable in vivo model to explore the involvement of microglia priming and activation in the development of cocaine addiction. Our findings showed that cocaine induced higher levels of locomotor activity and CPP, accompanied with higher activity of NLRP3 inflammasome in CX3CR1−/− mice than in WT counterparts. CX3CR1 deficiency could prime microglia and NLRP3 signaling under basal conditions. This study provided direct evidence supporting the two-hit hypothesis of microglia and the contributing roles of neuroinflammation in the development of cocaine addiction. Targeting the microglial NLRP3 pathway may provide an alternative approach for the beneficiary of cocaine addicts.

## 2. Methods and Materials

### 2.1. Animals and Reagents

CX3CR1-GFP (002582) and WT C57BL/6J mice were purchased from Jackson lab (Bar Harbor, ME, USA) and kept in the lab. All the animals were housed under conditions of constant temperature and humidity on a 12 h light, 12 h dark cycle, with lights on at 07:00 h. Food and water were available ad libitum. All animal procedures were performed according to the protocols approved by the Institutional Animal Care and Use Committee of the University of Nebraska Medical Center (project IACUC # 18-030-04-FC, and the most recent three-year renewal approval date 8 March 2021) and the National Institutes of Health. Cocaine hydrochloride (C5776) was purchased from Sigma and was freshly dissolved in 0.9% saline.

### 2.2. Behavioral Tests

**(A) Locomotor analysis**: CX3CR1−/− and WT mice (male, 3 months old, *n* = 6–8) were divided into four groups, receiving either cocaine (i.p., 20 mg/kg) or saline injections for 7 consecutive days: (1) WT mice administered saline; (2) WT administered cocaine; (3) CX3CR1−/− administered saline; and (4) CX3CR1−/− administered cocaine. On the 7th day, immediately following cocaine injection (i.p., 20 mg/kg), mice were put into the open field apparatus (Truescan, Coulbourn instrument) to detect locomotor activity for 30 min. The TruScan photobeam activity system consists of a clear arena with infrared sensors located on a ring 3 cm above floor level. There are 16 beams spaced 1 inch apart on the sensor ring that are used to detect movements by mice. These data are automatically relayed to a PC computer and interpreted by software. After the behavioral tests, mice were immediately sacrificed by CO_2_ euthanization for brain removal. Brain regions, including the striatum and hippocampus, were separated, flash frozen and stored at −80 °C for later use for protein and total RNA extraction. **(B) Conditioned place preference (CPP):** CX3CR1−/− and WT mice (3 months old, male, *n* = 6–8) were trained in a three-chamber apparatus (one central compartment and two side chambers), with one side of the chamber with white and black spots and the other side with grey stripes (Harvard apparatus, Holliston, MA, USA). On the pre-conditioning day, mice were put into a central compartment with free access to both chambers for 30 min (habituation). After a baseline preference measure, mice with no pre-existing bias for either side of the chamber were selected for the next step (time spent in either chamber <80% of total time). Mice were conditioned for 25 min/session over 2 days: to the cocaine-paired side (10 mg/kg i.p.) in the morning and to the saline-paired side in the afternoon, and an interval time was >6 h. On the fourth day, mice were placed in the central compartment and allowed to move freely between the two side chambers for 20 min. CPP scores were automatically calculated as time spent in the cocaine-paired chamber minus time spent in the saline-paired chamber by the connected computer (Harvard apparatus). The mice injected with saline (morning and afternoon) served as controls. The regimens we selected here (cocaine doses and injection days) for locomotor activity and CPP have been extensively employed by previous investigations and are well-accepted in the field of drug addiction [30,31,32,33].

### 2.3. Immunoblots

Frozen striatum or hippocampi were lysed in radioimmunoprecipitation assay (RIPA) buffer with 0.1% sodium dodecyl sulfate (SDS), protease and phosphatase inhibitors (Complete Mini, Roche, Basel, Switzerland) for 30 min, sonicated and clarified by centrifugation (1000× *g* for 10 min). Protein concentrations were determined using a bicinchoninic acid (BCA) Protein Assay Kit. Equal amounts of protein were loaded and run in sodium dodecyl-sulfate polyacrylamide gel electrophoresis (SDS-PAGE) gels, and subsequently transferred onto polyvinylidene fluoride (PVDF) membranes, followed by incubation with respective primary antibodies in 3% milk phosphate-buffered saline (PBS); 137 mM NaCl; 2.7 mM KCl; 10 mM Na_2_HPO_4_; 2 mM KH_2_PO_4_) overnight at 4 °C. Primary antibodies including NLRP3 (1:2000; AG-20B-0014), Caspase 1 (1:2000; AG-20B-0042) were purchased from adipogen (San Diego, CA, USA). Antibodies including ASC (1:1000; NBP1-78977), LC3II (1:3000; NB100-2220), CD11b (1:1000, NB110-89474), LAMP2 (1:1000, NB300-591), Iba-1 (1:1000, NB100-1028) were purchased from Novus Biologicals (Centennial, CO, USA); IL1β (1:2000; 65-0812R) was from Bioss (Woburn, MA, USA), Cathepsin D (1:1000, ab75852) was from Abcam (Waltham, MA, USA). Beclin1 (1:1000, sc-11427) was from Santa Cruz (Dallas, TX, USA), p62 (1:2000, PM045) was from MBL (Tokyo, Japan), GFAP (1:2000, G3893) was from Sigma (St. Louis, MO, USA). The next day, blots were incubated with peroxidase-conjugated secondary antibodies: goat anti-rabbit (1:3000, Santa Cruz, sc-2004) and goat anti-mouse (1:2500, Santa Cruz, sc-2005) for 1 h at room temperature. β-actin antibody (1:10,000; Sigma, A5441-2ML) was used for reprobing each membrane to ensure equal loading. Signals were detected using a chemiluminescent substrate for peroxidase (ECL, Pierce). Images were created using the ChemiDoc Imaging System (Bio-Rad) and analyzed using ImageJ software.

### 2.4. Immunofluorescence

Animals were perfused with 4% PFA and brains removed for histology. Rapidly frozen 30 µM brain sections were incubated with primary CD11b antibody (Bio-Rad, MCA711G) or GFAP, NLRP3 overnight at 4 °C. Secondary Alexa Fluor 488 goat anti-rabbit IgG (Invitrogen, A-11008) or Alexa Fluor 594 goat anti-mouse (Invitrogen, A-11032) was added for 2 h, followed by the mounting of sections with DAPI (Invitrogen, 3693). Fluorescent images were acquired at room temperature on a Zeiss Observer. A Z1 inverted microscope (Carl Zeiss, Oberkochen, Germany) was used; images were processed using AxioVs 40 4.8.0.0 software (Carl Zeiss MicroImaging, Oberkochen, Germany). Photographs were acquired using an AxioCam MRm digital camera (Carl Zeiss, Oberkochen, Germany).

### 2.5. RNA Extraction, Reverse Transcription, and Quantitative Polymerase Chain Reaction (qPCR)

Total RNA was extracted using Trizol reagent (Invitrogen, 15596-018). Briefly, 50–100 mg brain tissue was directly added to 1 mL Trizol. Brain lysates were briefly sonicated (3–5 s) and incubated for 10 min on ice then aspirated into new 1.5 mL microcentrifuge tubes with 0.2 mL of chloroform added. After extensive vortexing, the samples were centrifuged at 10,000× *g* for 15 min at 4 °C. The upper aqueous phase was transferred to a new tube and 500 µL isopropyl alcohol was added. Samples were incubated for 10 min and centrifuged again to precipitate total RNA. The total RNA was dissolved in DEPC-treated H2O and quantified. Reverse transcription reactions were performed using a Verso cDNA kit (Invitrogen, AB-1453/B). The reaction system (20 µL) included 4 µL 5X cDNA synthesis buffer, 2 µL dNTP mix, 1 µL RNA primer, 1 µL RT enhancer, 1 µL Verso enzyme Mix (Invitrogen, AB-1453/B), total RNA template 1 µg, and a variable volume of water. Reaction conditions were set at 42 °C for 30 min. QPCRs were performed by using SYBR Green ROX qPCR Mastermix (Qiagen, 330510, Hilden, Germany). Reaction systems were set up as follows: 10 µL SYBR Green Mastermix, 0.5 µL forward primers, 0.5 µL reverse primers, and 9 µL distilled H2O. A number of 96-well plates were placed into a 7500 fast real-time PCR system (Applied Biosystems, Grand Island, NY, USA) for program running. Mouse primers for Tnf, Il6, Il1b, Ccl2, TGFb, IL4 were purchased from Invitrogen, Mm00443258, Mm00446190, Mm00434200, and Mm00441242, Mm00493650, Mm00445259.

### 2.6. IL1β ELISA

Protein homogenates of the striatum or hippocampus were isolated the brains of WT and CX3CR1−/− mice and assessed for expression of IL1β protein levels using a commercially available ELISA kit (R&D Systems, MLB00C).

### 2.7. Statistics

The results are presented as means ± SEM. For comparisons between two groups, an unpaired two-tailed student’s T-test was used. For comparisons between multiple groups, one-way ANOVAs followed by Bonferroni post hoc tests were employed. All statistical tests were performed with GraphPad Prism (La Jolla, CA, USA). Probability levels of < 0.05 were considered statistically significant. A minimum of four biological replicates were used for all experiments.

## 3. Results

### 3.1. Enhanced Cocaine-Mediated Locomotor Activity, CPP, and NLRP3 Inflammasome Activity in CX3CR1-Deficient Mice

To explore whether CX3CR1 deficiency could regulate cocaine-mediated hyperlocomotion, CX3CR1−/− and WT mice were administered with cocaine (i.p., 20 mg/kg) or saline for 7 days. On the seventh day, immediately after the cocaine/saline injection, the mice were assessed for their locomotor activities. On the next day, all mice were sacrificed by CO_2_ euthanization for brain removal. While there was no significant difference with saline administration between the two groups of mice, in the presence of repeated cocaine administration, CX3CR1−/− mice exhibited increased locomotor activity compared to the WT mice (*p* < 0.05; Figure 1A). We did not observe a significant difference in locomotor activities between these two groups on the first day (Appendix A). We then performed CPP to determine the regulation of CX3CR1 deficiency on cocaine-mediated “motivational effects”. Mice were administered with cocaine (10 mg/kg, I.P.) and restrained in one chamber with dots for 25 min (morning), or saline and restrained in another chamber with stripe for 30 min (afternoon) for two consecutive days. The following day, mice were put into the middle of the apparatus and allowed to freely explore the different chambers. The times in saline- or cocaine-associated chambers were automatically calculated by the connected computer. Our results showed that CX3CR1−/− mice spent longer times in the cocaine-paired environment compared with the WT controls (*p* < 0.05; Figure 1B). These results indicated that cocaine induced a higher degree of reward effects in CX3CR-deficient mice than in WT mice. Previous studies showed that IL1β is a critical mediator involved in LPS-mediated increased activation of microglia in CX3CR1−/− mice and the changes in IL1β levels are involved in addiction [7,28]. We next sought to examine the total amount of IL1β in the brains of mice with 7-day cocaine administration by the ELISA approach. We found that cocaine could increase the total IL1β amount in the striatum but not in the hippocampus of the CX3CR1−/− mice compared with the WT controls (Figure 1C). Since ELISA detects both pro- and mature (m) IL1β, we next examined the levels of mIL1β in the brains of mice with cocaine exposure. We found significantly increased expression of mIL1β in both the striatum and hippocampus of CX3CR1−/− mice compared to WT controls (* *p* < 0.05, Figure 1D–F). Increased levels of mIL1β are mainly derived by NLRP3 inflammasome activation [34]. We then assessed the status of NLRP3 in the these brains of mice. As shown in Figure 1G,H, in the presence of cocaine, CX3CR1−/− mice revealed significant upregulation of NLRP3 (* *p* < 0.05) compared to the WT controls, suggesting the NLRP3 inflammasome was indeed involved in cocaine-induced production of mIL1β in CX3CR1−/− mice. The increased mIL1β levels could be either derived from the increased NLPR3 inflammasome activity or from increased transcription. To discern this, we detected the mRNA levels of IL1β in WT and CX3CR1−/− mice with saline and cocaine injections. As shown in Appendix A, CX3CR1−/− mice revealed increased mRNA levels in the striatum compared to WT mice under saline injections (lane 2 vs. lane1, * *p* < 0.05), while cocaine could not further increase IL1β mRNA levels (*p* > 0.05, lane 4 vs. lane 2). These results strongly suggested that the increased mILβ production was probably derived from increased activity of NLRP3 inflammasome, and CX3CR1 deficiency could prime NLRP3 signaling by increasing IL1β mRNA levels. We observed the same upregulation trend in mRNA levels of IL6, ccl2, and TNFα in the brains of these four groups of mice (Appendix A). We also found that TGFβ was upregulated in CX3CR1-deficient mice under basal levels, but this upregulation was blocked by cocaine administration (Appendix A). We did not find any changes on another anti-inflammatory factor IL4 on these four groups of mice (*p* > 0.05, Appendix A).

CX3CR1−/− mice showed increased activity in the canonical NLRP3 inflammasome pathway and increased lysosomal markers with cocaine administration. We have shown increased NLRP3 levels in the brains of CX3CR1−/− mice under repeated cocaine administration; the next aim was to explore the status of other components of NLRP3 inflammasomes, including apoptosis-associated speck-like protein containing a CARD (ASC) and caspase 1. The results showed that cocaine resulted in an increased trend of ASC oligomerization in both the striatum and hippocampus but did not reach significances (*p* > 0.05, Figure 2A,B). However, we found a significant upregulation in mature (m) caspase 1 (activate form) in the striatum and hippocampus of CX3CR1−/− mice compare to WT mice (* *p* < 0.05, Figure 2C,D). There was no difference in maturation of pro-caspase-11 in the brains of CX3CR1−/− and WT mice (non-canonical NLRP3 pathway, data not shown). Previous studies have shown that cocaine increased the cathepsin B secretion from the lysosome in HIV (+) individuals [35] and have deleterious effects on lysosome degradation function [36]. CX3CR1 deficiency could also induce dysregulation in the autophagy–lysosome degradation pathway and impair the autophagosome maturation process in mice with amyotrophic lateral sclerosis [37]. Since lysosomal damage and the released cathepsins could contribute to NLRP3 activation through cathepsin–NLRP3 binding as the second signal [38,39], we next sought to explore lysosome function status in WT and CX3CR1−/− mice with cocaine administration. CX3CR1−/− mice revealed significantly higher levels of the glycosylated form of lysosomal-associated membrane protein 2 (LAMP2) in the striatum (* *p* < 0.05, Figure 3A) and both the native and glycosylated forms in the hippocampi (* *p* < 0.05, Figure 3C). These results indicated there was more lysosome accumulation in the brains of CX3CR1−/− mice. We then checked the levels of lysosomal cathepsin D in the brains of these two groups. CX3CR1−/− mice showed significant upregulation of mature levels of cathepsin D in both the striatum and hippocampi compared to WT mice (* *p* < 0.05, Figure 3E,G). These results indicated that lysosomes were more dysregulated in CX3CR1−/− mice than in WT mice with cocaine administration. Lysosome function is closely linked to autophagy flux, we then explored whether lysosome dysfunction could affect the autophagy process in CX3CR1−/− mice., we did not observe a significant difference in the levels of beclin1, p62, and LC3II, which indicated that there was not much difference in autophagy flux between WT and CX3CR1 −/− mice with cocaine administration (Appendix A, *p* > 0.05). Taken together, our findings implied that the increased NLRP3 inflammasome activity in CX3CR1−/− mice with cocaine was probably derived from the upregulation of the canonical NLRP3 inflammasome signaling and dysregulated lysosomal function.

### 3.2. CX3CR1 Deficiency-Primed NLRP3 Inflammasome Signaling under Basal Conditions

CX3CR1−/− mice showed increased mRNA levels of IL1β, TNFα, and IL6 in the brain compared to WT mice under saline injections (Appendix A), suggesting that CX3CR1−/− could activate or prime glial cells. To validate this, we determined the levels of CD11b and GFAP in the striatum of CX3CR1−/− and WT mouse brains. The results from both Western blots and immunofluorescence showed there were significant higher levels of CD11b and GFAP levels in CX3CR1−/− mice than in WT mice (Appendix A). Then, we explored the effects of CX3CR1 deficiency on NLRP3 inflammasome signaling in the brain under basal conditions. Interestingly, we found that CX3CR1 deficiency increased the levels of NLRP3 in the striatum (Figure 4A and Appendix A), and further validation was obtained by immunostaining, showing increased NLRP3 intensity in CX3CR1−/− mice (Appendix A). Additionally, NLRP3 was shown in the microglia, revealed by the co-localization of GFP and NLRP3 signals (Appendix A). We then explored the status of ASC, caspase 1, and IL1β in these two groups of mice under basal conditions. We found that ASC oligomerization was significantly increased in the brains of CX3CR1−/− mice (Figure 4A,B and Appendix A) but not for the mature caspase 1 and mIL1β production (Figure 4C–F and Appendix A). Taken together, these results revealed that CX3CR1 deficiency could increase IL1β mRNA levels, NLRP3 protein levels, and ASC oligomerization, but not for caspase 1 activation and mature IL1β production. These findings suggested that CX3CR1−/− could prime the NLRP3 inflammasome.

### 3.3. CX3CR1 Deficiency Increased Lysosome Biogenesis under Basal Condition

We then investigated whether CX3CR1 deficiency could also affect lysosomal function. As shown in Figure 5A,B, the levels of glycosylated LAMP2 were significantly increased in both striatum and hippocampus of CX3CR1−/− mice compared to WT mice (Appendix A, * *p* < 0.05). Additionally, the levels of pro- and mature cathepsin D was also significantly increased in the striatum but not in the hippocampus of CX3CR1−/− mice (*p* < 0.05, Figure 5C,D). These results suggested that CX3CR1−/− deficiency could affect lysosomal biogenesis and mature process.

## 4. Discussion

In this study, we employed CX3CR1−/− mice to demonstrate that microglia priming could significantly increase cocaine-mediated reward effects, and NLRP3 inflammasome signaling and lysosome function were involved in microglia prime. Our findings (listed in Appendix A) were consistent with previous findings and provided more solid evidence indicating the roles of microglia in drug addiction.

Microglia, the brain residential macrophages. constitute the first line of defense to the senses, respond to both internal and external insults, and function as housekeepers, guards, and warriors to maintain brain homeostasis in physiological and pathological conditions [16]. Instead of the previously mentioned two functional statuses called the resting and activation statuses, microglia are now believed to be in a continuum status, ranging from resting state to full activation. The concepts of “microglia priming” have been proposed to explain the exaggerated immune responses when microglia receive a second insult [40,41,42,43]. In prime status, microglia could not be differentiated from the resting state by morphological analysis or even at transcriptional levels. However, primed microglia are very sensitive and reveal much stronger immune responses on receiving insults than non-primed microglia do. Several mechanisms have been suggested to explain the priming status of microglia, including the changes in chromatin condense state, methylation status on the promoter regions of certain genes, and the alterations in microRNA levels [40,44]. Recently, microglia priming has been suggested to play critical roles in aging, traumatic CNS injury, and the development of neurodegenerative disease [45,46].

To explore whether microglia priming could also play contributing roles in the development of drug addiction, we compared the reward effects between CX3CR1−/− and WT mice under different cocaine regimens. Our results demonstrated that CX3CR1−/− mice showed increased cocaine-mediated locomotor activity under 7-day repeated injections and CPP score compared to WT mice, indicating that CX3CR1−/− mice have enhanced reward effects. We did not observe the basal levels of locomotor activity between CX3CR1−/− and WT mice, probably as CX3CR1 deficiency can only prime microglia by increasing mRNA levels of IL1β. At such conditions, NLRP3 inflammasome was primed but not activated, and the basal levels of neuron excitability between CX3CR1−/− and WT mice have no significant differences. Therefore, NLRP3 needs the second insult (cocaine) to produce the mature form of IL1β which could act on neurons, leading to the increased excitability of neurons. We also did not observe any significant difference in locomotor activity between CX3CR1−/− and WT mice under the first cocaine injection (Appendix A). The mechanisms underlying this phenomenon are not clear. It is possible that only one injection of cocaine is not strong enough to induce NLRP3 full activation. We also understand that locomotor activity and seeking behavior represent only certain aspects of drug addiction. To confirm the roles of microglia priming and NLRP3 inflammasome in the development of drug addiction, investigations using a cocaine self-administration model are guaranteed in future studies.

In this study, we selected two brain regions, the striatum and hippocampus, for biochemical analyses. The rationales are: (1) the striatum and hippocampus are the critical components of brain-reward circuitry and are very sensitive to abused drugs at the initial exposure time period; (2) the striatum and hippocampus are closely related to cocaine-mediated hyper-locomotor activity and CPP, respectively. We will investigate the changes in other brain regions in later studies. We found that CX3CR1−/− mice revealed higher NLRP3 inflammasome activity and higher levels of lysosomal cathepsin D with cocaine injections. Under basal levels, CX3CR1 deficiency increased the mRNA levels of IL1β, protein levels of NLRP3, and also upregulated LAMP2 glycosylation levels. Our data were consistent with previous results showing that increased lysosomal activity was identified in CX3CR1−/− mice [47,48]. It is well-known that NLRP3 inflammasome activation needs two signals: the first one is to prime NLRP3 by increasing the transcriptional levels of IL1β and NLRP3 and the second one is to increase NLPR3 inflammasome formation, resulting in enhanced production of mature IL1β [18,19]. Our data implied that CX3CR1 deficiency could act as the first signal to prime NLRP3, and cocaine served as the second one, leading to the full activation of NLRP3, evidenced by the increased levels of mIL1β. We still did not know how cocaine upregulated NLRP3 inflammasome activity. However, we observed that increased lysosome biogenesis and numbers in the brain occurred due to CX3CR1 deficiency; it is possible that cocaine could disrupt the membrane of the lysosome, leading to an increased release of cathepsins, which could directly interact with NLRP3, a known second signal for NLRP3 inflammasome activation [38,39]. Previous studies have implied that cocaine was capable of interfering with lysosomal membrane permeability, resulting in the release of cathepsins into the cytoplasm in certain conditions [35,36]. Still, the detailed mechanisms underlying how cocaine affected lysosomal function warrants more investigations in the future study. Additionally, the effects of NLRP3 inhibition on cocaine-mediated behavioral changes in CX3CR1−/− mice are needed to further corroborate our hypothesis.

Previous epidemiological studies have revealed that ELS, including brain trauma, mental and physical abuse, family violence, child maltreatment, etc., could increase the likelihood of drug addiction in later life [49,50]. Increased levels of peripheral and neural inflammation in ELS-experienced individuals were implied to play critical roles in the development of neuropsychiatric and/or neurodegenerative diseases [51]. Numerous rodent models showed that ELS was capable of having a long-term effect on microglial functional status. For example, ELS could increase microglial engulfment of neuronal spine to induce depressive symptoms in late adolescence [52]. ELS could also worsen adult remote microglia activation, neuronal death, and functional recovery after focal brain injury in mice [53] and alter emotional behaviors in rats [54]. Overall, these studies indicated that ELS could prime microglia, leading to exaggerated microglial immune responses in later life, which is responsible for various neurological disorders. The concepts of “microglia priming” and the “two-hit hypothesis” have guided work in the field of neuroimmunology over the last decade [40,41,42,43]. Microglia are highly plastic and could be pre-activated or ‘primed’ by an earlier inflammatory event, which leads to amplified responses to a second inflammatory insult, that is, “innate immune memory (IIM).” Our data indicated that NLRP3 signaling priming and activation was responsible for microglia priming and activation and bridged the linkage between ELS and drug addiction. Consistently, ELS exposure increased expression of NLRP3 inflammasome proteins and anxiety-like behavior in adolescent rats [55], and NLRP3 inflammasome inhibition mitigated ELS-induced cognitive impairment in adult mice [56].

The CX3CR1/CX3CL1 axis is critical to maintain microglia in a quiescent state and is very sensitive to various types of stimuli (stress). Therefore, CX3CR1-deficient mice have been extensively employed to explore the roles of microglia in various psychiatric diseases. For example, chronic mild stress (CMS) could decrease CX3CR1 levels in the hippocampus, and this change was associated with CMS-mediated microglial activation and depression-related behaviors [57]. In addition, CX3CR1-deficient (CX3CR1−/−) mice elicited increased fear acquisition as well as reinstatement of fear, which are related to post-traumatic stress disorders [58]. Mechanistically, CX3CR1-deficient mice showed increased neuronal activity compared to WT mice following the second insults [58]. Accumulating evidence indicates that drug addiction is also a neuroimmune-related brain diseas and the crosstalk between microglia and neurons is critical for promoting drug addiction. So, using CX3CR1-deficient mice is a suitable model to investigate the involvement of microglia in the development of drug addiction.

Our findings provided a novel mechanism to possibly explain the positive association between ELS and drug addiction. The limitation for this work is that CX3CR1 deficiency is a lifelong condition, while early life stresses (ELS) are usually transient events. However, the effects of ELS on microglia function and neuroinflammation are long-term effects. Several rodent models have demonstrated that ELS could impact the development of microglia and dysregulate microglia functional and increase neuroinflammation levels when the rodents reach adolescence and adulthood [55,59,60,61]. Another limitation is that we focused on microglial NLRP3 only. Actually, NLRP3 inflammasome is also expressed in other brain cells including astrocytes [62,63], and astrocyte activation also contributes to the elevation of neuroinflammation levels. The astrocyte NLRP3 inflammasome might also participate in the potentiation of cocaine-mediated reward effects in CX3CR1 deficiency mice.

In summary, our findings have demonstrated that CX3CR1−/− mice showed increased sensitivity to cocaine-mediated behavioral changes compared to the WT mice, and furthermore, that the NLRP3-IL1β axis plays a key role in this process. We hypothesized that CX3CR1 deficiency first primed the NLRP3 inflammasome by augmenting the expression levels of NLRP3 and that cocaine could possibly potentiate NLRP3 inflammasome activity by affecting lysosomes. Increased mIL1β could, in turn, impact neuronal excitability, leading to the exacerbation of cocaine-mediated behavioral phenotypes. Our findings agree with previous reports implicating a link between psychological symptomatology and upregulated IL1β in the brains of cocaine addicts [64]. Our findings showing the involvement of the NLRP3 inflammasome in drug addiction have a broader biological significance. They provided a possible explanation that an initial stimulus such as stress or traumatic injury that primed microglia (NLRP3) can lead to increased likelihood for drug addiction later in life [65,66,67,68]. The regulation of microglial activation status via targeting of the NLRP3-IL1β axis could open new avenues for preventing future addictive behaviors.

## 5. Limitation Section

We understand that we have some limitations in our research. For biochemical analysis, it would be better to run all four groups (CX3CR1 deficiency and WT mice with cocaine or saline injections) together and analyze the results together using two-factor ANOVA (treatment X genetic variance) to obtain more convincing results. Additionally, another group without any injection can be added to control “injection effects”. For CPP experiments, we obtained the CPP score by calculating the time difference (the time difference between the saline-paired environment and cocaine-paired environment) of each mouse in the testing day, and then compared and analyzed whether the scores between these two strains of mice have significant differences. This approach has been the most widely used and accepted in CPP tests. We acknowledged that both WT and CX3CR1-deficient mice did not reach the bias to the cocaine-paired environment (>80% of total time period) in our regimen. Additionally, for a CPP regimen it would be better to include a counterbalance (switching the time for cocaine/saline injections) approach to avoid time-dependent effects.

## Figures and Tables

**Figure 1 jpm-11-00963-f001:**
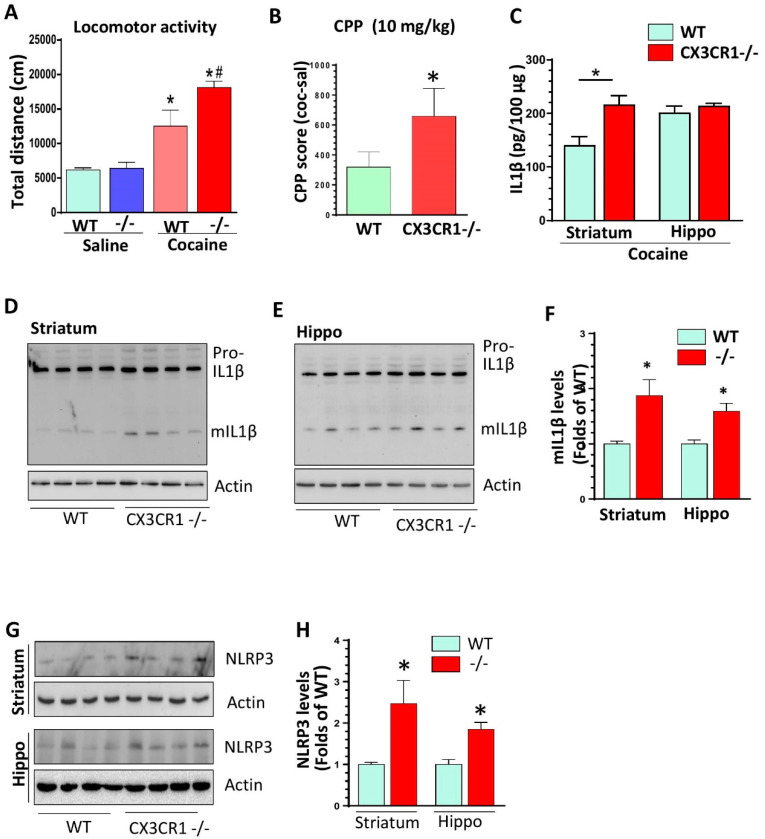
CX3CR1−/− mice showed potentiation of cocaine-induced behavioral changes and enhanced expression levels of mature IL1β in the brains compared to wild type (WT) controls. (**A**) Repeated 7-day cocaine injections induced increased locomotor activities in CX3CR1−/− mice compared to WT controls (*n* = 6–8; one-way ANOVA analysis. * *p* < 0.05, group of WT with cocaine vs. group of WT with saline; # *p* < 0.05, group of Cx3CR1−/− with cocaine vs. group of WT with cocaine). (**B**) Cocaine induced higher CPP in CX3CR1−/− mice compared with WT controls. (*n* = 6–8, * *p* < 0.05; t-student analysis). (**C**) ELISA results showed that CX3CR1−/− mice had higher levels of total IL1β in the striatum but not in the hippocampus following repeated cocaine injections. (*n* = 6–8, * *p* < 0.05; CX3CR1−/− vs. WT; t-student analysis). (**D**–**F**) CX3CR1−/− mice showed increased mIL1β levels in the striatum and hippocampus compared to WT mice with repeated cocaine injections (*n* = 6–8, * *p* > 0.05; CX3CR1−/− vs. WT; t-student analysis). (**G**,**H**) CX3CR1−/− mice showed increased NLRP3 protein levels in the striatum and hippocampus compared to WT mice with repeated cocaine injections (*n* = 6–8, * *p* < 0.05; CX3CR1−/− vs. WT; *t*-student analysis).

**Figure 2 jpm-11-00963-f002:**
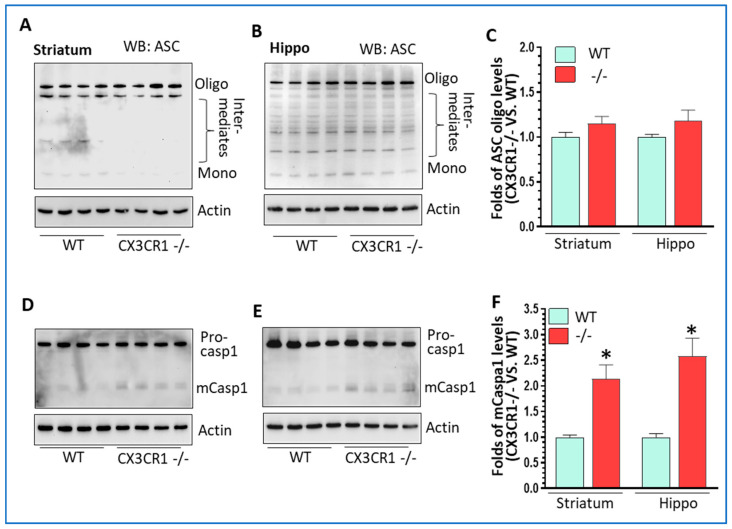
The levels of NLRP3 inflammasome in cocaine-treated WT and CX3CR1−/− mice. (**A**) Representative WBs showed no difference in ASC oligomers in the striatum of CX3CR1−/− mice compared to WT mice with cocaine administration (*n* = 4–6, *p* > 0.05). (**B**) Representative WBs showed no difference in ASC oligomers in the hippocampus of CX3CR1−/− mice compared to WT mice with cocaine (*n* = 4–6, *p* > 0.05). (**C**) Statistical results of ASC oligo WBs from A and B. (**D**) Representative WBs showed a significant increase in mature caspase 1 in the striatum of CX3CR1−/− mice compared to WT mice with cocaine (*n* = 4–6, * *p* < 0.05). (**E**) Representative WBs showed a significant increase in mature caspase 1 in the hippocampus of CX3CR1−/− mice compared to WT mice with cocaine (*n* = 4–6, * *p* < 0.05). (**F**) The statistical results of mCasp 1WBs from (**D**,**E**).

**Figure 3 jpm-11-00963-f003:**
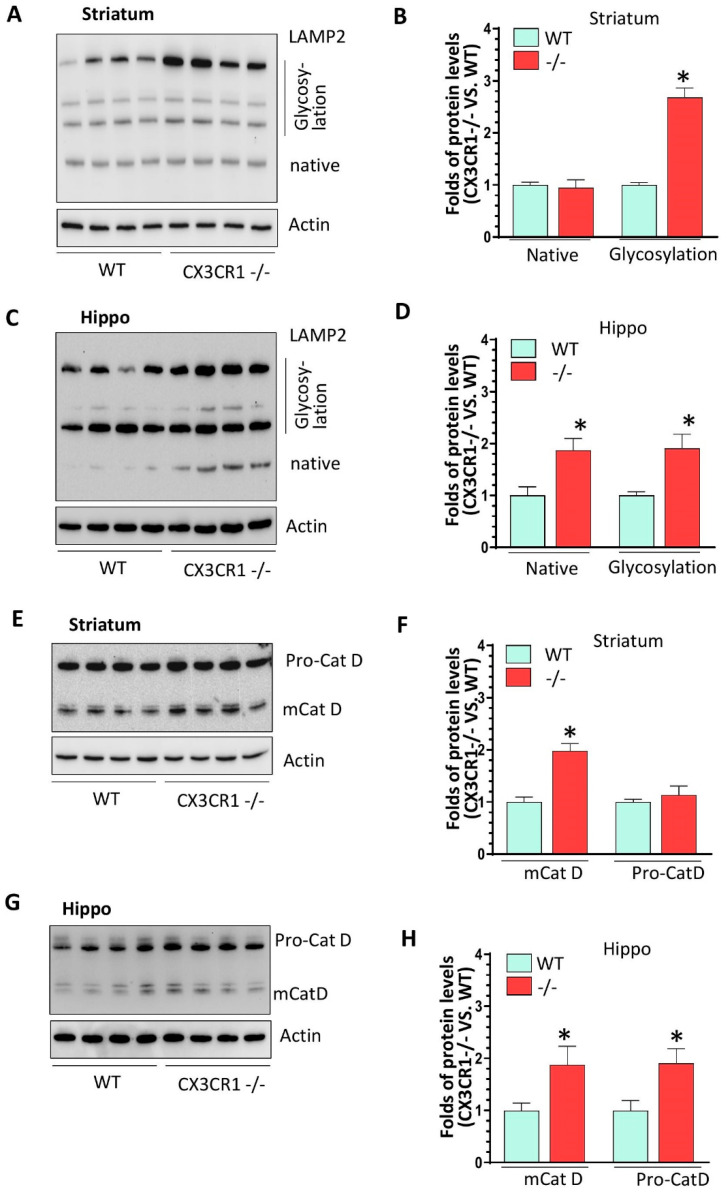
The levels of lysosomal markers in cocaine-treated WT and CX3CR1−/− mice. (**A**,**B**) Representative WBs showed increased levels of LAMP2 glycosylated form but not for its native form in the striatum of CX3CR1−/− mice compared to WT mice with cocaine administration (*n* = 4–6, * *p* < 0.05). (**C**,**D**) Representative WBs showed increased levels of LAMP2 native and glycosylated form in the hippocampus of CX3CR1−/− mice compared to WT mice with cocaine administration (*n* = 4–6, * *p* < 0.05). (**E**,**F**) Representative WBs showed increased levels of mature Cat D in the striatum of CX3CR1−/− mice compared to WT mice with cocaine administration (*n* = 4–6, * *p* < 0.05). (**G**,**H**) Representative WBs showed increased levels of both pro- and mature Cat D in the hippocampus of CX3CR1−/− mice compared to WT mice with cocaine administration (*n* = 4–6, * *p* < 0.05).

**Figure 4 jpm-11-00963-f004:**
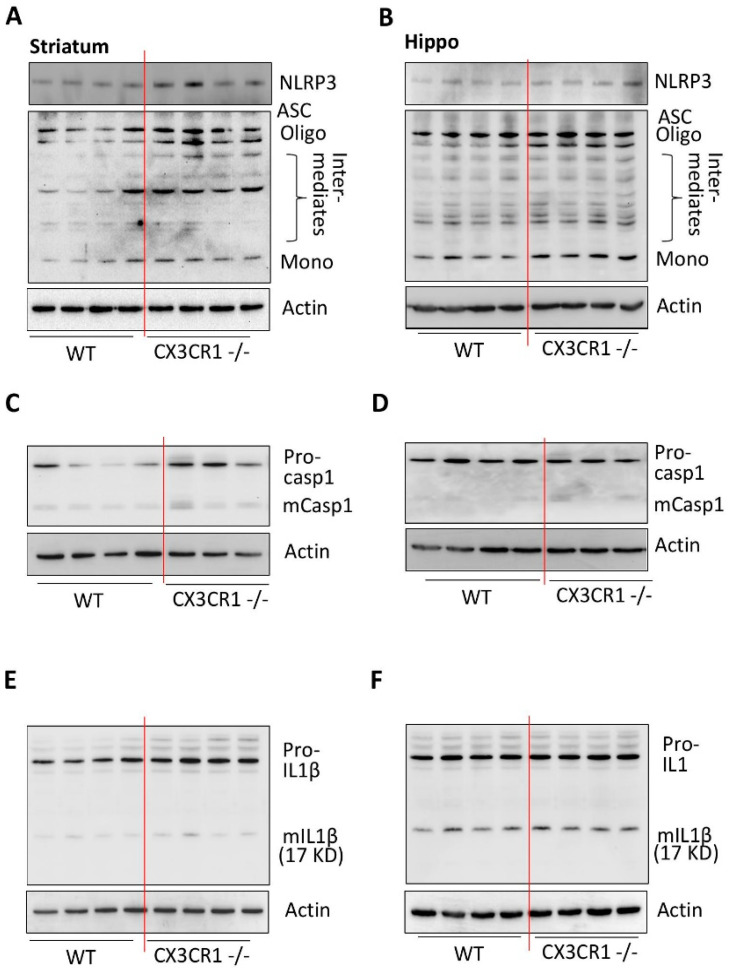
CX3CR1 deficiency primes NLRP3 inflammasome by increasing NLRP3 protein levels and ASC in the striatum compared to WT mice at basal conditions. (**A**) Representative WBs showed increased levels of NLRP3 and increased ASC oligomerization in the striatum of CX3CR1−/− mice compared to WT controls at basal conditions (*n* = 4–6). (**B**) Representative WBs showed no increased levels of NLRP3 but increased ASC oligomerization in the hippocampus of CX3CR1−/− mice compared to WT controls at basal conditions (*n* = 4–6, *p* < 0.05). (**C**) Representative WBs showed no increased levels of mature caspase 1 in the striatum of CX3CR1−/− mice compared to WT controls at basal conditions (*n* = 3–6, *p* < 0.05). (**D**) Representative WBs showed no increased levels of mature caspase 1 in the hippocampus of CX3CR1−/− mice compared to WT controls at basal conditions (*n* = 3–6, * *p* < 0.05). (**E**) Representative WBs showed no increased levels of mature IL1β in the striatum of CX3CR1−/− mice compared to WT controls at basal conditions (*n* = 4–6, *p* > 0.05). (**F**) Representative WBs showed no increased levels of mature IL1β in the hippocampus of CX3CR1−/− mice compared to WT controls at basal conditions (*n* = 4–6, *p* > 0.05).

**Figure 5 jpm-11-00963-f005:**
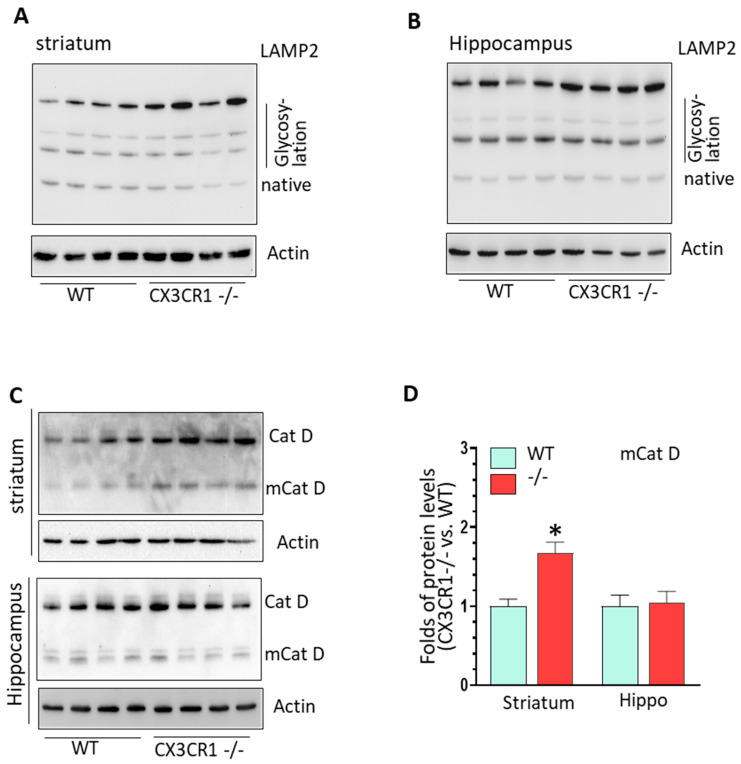
CX3CR1 deficiency regulated lysosomal biogenesis process in the brain at basal conditions. (**A**) Representative WBs showed increased levels of LAMP2 glycosylated form in the striatum of CX3CR1−/− mice compared to WT controls under basal levels (*n* = 4–6, * *p* < 0.05). (**B**) Representative WBs showed increased levels of LAMP2 glycosylated form in the hippocampus of CX3CR1−/− mice compared to WT controls at basal levels (*n* = 4–6, * *p* < 0.05). (**C**) Representative WBs showed increased levels of mature cathepsin D in the striatum but not in hippocampus of CX3CR1−/− mice compared to WT controls under basal levels (*n* = 4–6, * *p* < 0.05). (**D**) Statistical analysis for mature Cat D WBs shown in (**C**).

## Data Availability

He data presented in this study are available in article or supplementary material here.

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
