# Peer review of "NLRP3 Inflammasome Is Involved in Cocaine-Mediated Potentiation on Behavioral Changes in CX3CR1-Deficient Mice"

_jpm, 2021, doi:10.3390/jpm11100963_

Round 1
Reviewer 1 Report
The article prepared by Guo et al entitled “NLRP3 inflammasome is involved in cocaine-mediated potentiation on behavioral changes in CX3CR1 deficient mice” described the expression of IL1beta and inflammation-related molecules in CX3CR1 KO and WT mice with or without repetitive cocaine treatment. The results are consistent. However, there are several critical issues needed to be resolved.
Major concerns
- The effects of CX3CR1 deficiency on IL1beta and inflammation-related molecules were not quantified and justified. The authors should first compare the quantitative results between WT and CX3CR1-/- before showing the biochemical effects of repetitive cocaine treatments.
- The authors employed CX3CR1-GFP (CX3CR1-/-), a genetic deficiency, as early life stress and proposed cocaine treatment as a second hit to evaluate the role of IL1beta and inflammation-related molecules in the two-hit hypothesis of drug addiction. The authors have to explain how CX3CR1 deficiency could simulate the condition of early life stress. The discrepancy between CX3CR1 deficiency, a lifelong condition, and early life stress, usually transient events, should be considered.
- The authors should provide the rationales of why examining the molecular expressions in the striatum and hippocampus, but not other brain regions. Besides, the authors should discuss how the changes in CX3CR1-/- mice associate with unchanged locomotor activity following saline and first cocaine treatment.
- The results of CPP in this study demonstrated that CX3CR1-/- mice exhibit a greater preference toward cocaine compared with WT mice. However, the authors seemed to take this as a sign of drug addiction. This conceptual gap should be pointed out.
Minor concerns
- In the figure legends, the description could be shortened. Repeated statements should be avoided. The usage of beta-actin could be stated in the materials and methods.
- CX3CR1-/- mice is a condition, not a treatment. The title of figure 2 “CX3CR1-/- mice upregulated NLRP3 inflammasome activity….” might be stated as “The levels NLRP3 inflammasome in cocaine-treated WT and CX3CR1-/- mice”.
Author Response
Reviewer 1:
The article prepared by Guo et al entitled “NLRP3 inflammasome is involved in cocaine-mediated potentiation on behavioral changes in CX3CR1 deficient mice” described the expression of IL1beta and inflammation-related molecules in CX3CR1 KO and WT mice with or without repetitive cocaine treatment. The results are consistent. However, there are several critical issues needed to be resolved.
Major concerns
- The effects of CX3CR1 deficiency on IL1beta and inflammation-related molecules were not quantified and justified. The authors should first compare the quantitative results between WT and CX3CR1-/- before showing the biochemical effects of repetitive cocaine treatments.
Response: Great point! Actually, we compared the mRNA levels of IL1beta, IL6, CCL2, TNFα, and TGFβ in CX3CR1 deficiency and WT mice without repeated cocaine administration (suppl. Fig. 1) and the results showed that CX3CR1 deficiency could increase mRNA levels of IL1beta and inflammation-related molecules.
- The authors employed CX3CR1-GFP (CX3CR1-/-), a genetic deficiency, as early life stress and proposed cocaine treatment as a second hit to evaluate the role of IL1beta and inflammation-related molecules in the two-hit hypothesis of drug addiction. The authors have to explain how CX3CR1 deficiency could simulate the condition of early life stress. The discrepancy between CX3CR1 deficiency, a lifelong condition, and early life stress, usually transient events, should be considered.
Response: We agree with the reviewer’s point! CX3CR1 deficiency is a lifelong condition while early life stresses (ELS) are usually transient event. We acknowledged the difference between those two conditions. However, the effects of ELS on microglia function and neuroinflammation are long-term effects. Several rodent models have demonstrated that ELS could impact the development of microglia and dysregulate microglia functional and increase neuroinflammation levels when the rodents reach adolescence and adulthood [1-4]. So, we argue that CX3CR1 deficiency could mimic the effects of ELS. Such explanation has been added into the discussion section.
3. The authors should provide the rationales of why examining the molecular expressions in the striatum and hippocampus, but not other brain regions. Besides, the authors should discuss how the changes in CX3CR1-/- mice associate with unchanged locomotor activity following saline and first cocaine treatment.
Response: Yes. The rationales for choosing these two brain regions are: (1) the striatum and hippocampus are the critical components of brain reward circuitry and very sensitive to abused drugs at initial exposure time period; (2) the striatum and hippocampus are closely relevant to cocaine-mediated hyper-locomotor activity and CPP, respectively. So, we selected and investigated the possible changes in these two brain regions in our studies. We will investigate other brain regions in later studies. For behavioral studies, we did not observe the basal levels of locomotor activity between CX3CR1 -/- and WT mice, probably CX3CR1 deficiency can only prime microglia by increasing mRNA levels of IL1β. At such condition, NLRP3 inflammasome was primed but not activated and the basal levels of neuron excitability between CX3CR1 -/- and WT mice have no significantly difference. NLRP3 needs the second insult (cocaine) to produce the mature form of IL1β, which could act on neurons leading to increased excitability of neurons. We also did not observe the significant difference on locomotor between CX3CR1 -/- and WT mice under the first cocaine injection. We are not clear the mechanisms underlying this phenomenon. It is possible that only one injection of cocaine is not strong enough to induce NLRP3 full activation. Such discussions have been added into the discussion section.
- The results of CPP in this study demonstrated that CX3CR1-/- mice exhibit a greater preference toward cocaine compared with WT mice. However, the authors seemed to take this as a sign of drug addiction. This conceptual gap should be pointed out.
Response: Yes. CPP is sign of “seeking effects” of abused drug. The conceptual gap between seeking effects and drug addiction has been added into the discussion section.
Minor concerns
- In the figure legends, the description could be shortened. Repeated statements should be avoided. The usage of beta-actin could be stated in the materials and methods.
Response: Yes. We shortened the figure legends and removed the repeated statements for beta-actin. The usage of beta-actin was stated in the Material and methods part.
- CX3CR1-/- mice is a condition, not a treatment. The title of figure 2 “CX3CR1-/- mice upregulated NLRP3 inflammasome activity….” might be stated as “The levels NLRP3 inflammasome in cocaine-treated WT and CX3CR1-/- mice”.
Response: Yes. We changed the title of figure 2 as well as the figure 3.
Reviewer 2 Report
The manuscript entitled “NLRP3 inflammasome is involved in cocaine-mediated potenti-ation on behavioral changes in CX3CR1 deficient mice” By Guo et al., described really interesting studies on the role of innate immune states on some cocaine addiction induced symptoms and markers of inflammation. All together the sudy present a huge range of inflammation marker analysis which is very complete, and of very good interest for the field, but does address as much as they could the effect of either cocaine or the CX3CR1 gene deletion. Some consequences to other brain cells known to be very sensitive to inflammation could be put in light/perspectives.
Concerns:
- Introduction: the direct relationship with ELS is perhaps premature, and could be addressed in the discussion section but not here. However further precisions on the innate immune system could be address in order for the reader to better understand the choices for biochemical analysis (for example gene expressions, microglial phenotypes, apoptosis)…
- Material and methods: the amount of time between the last-injection and/or the last behavioural experiment and the euthanasia is never addressed and should be. Justify why the cocaine dose and the number of injections is different with locomotor analysis and CPP tests. Therefore, we do not know what means “animals under cocaine” in each biochemical analysis, (how many injections? Dose? Days or hours between the last injection and the measurement?)
- CPP experiments are measuring “drug-seeking behaviour” instead of “wanting effects” page 5 line 189. If there is experiments that includes some saline-treated animals, that is not the case for CPP experiments, why? Some controls are needed. In the same manner, there is no “cocaine result in”… you could not say something like that : there is no saline controls in any of the biochemical data in graphs of (figure 1, 2, 3)… The only figure 4 is presented without graphical representations to bring the assurance that there is no effects on some inflammation markers between WT and CX3CR1-/-. This is not convincing. Perhaps a two factor ANOVA (one factor treatment, one factor genotype), could be better for each structure, as the striatum and the hippocampus can be separated in the analysis and graphical representations. We need to know exactly which are the cocaine effects and which are only due to genotype and which are both…
- Figure 2: missing (E) in the legend, and there is two (C) and the graph in (C) is not in the legend also. Figure 5: I do not understand the (D) point.
- Results/Discussion: The interleukines mRNA expression are evaluated in the brain, so we do not know for sure which is the kind of cell that increased its IL1 expression, be careful in your conclusions. Is there any difference or effect in living neurons or oligodendrocytes in the hippocampus and striatum following this huge inflammation after cocaine administration and or CX3CR1 deletion?
Author Response
Reviewer 2:
Comments and Suggestions for Authors
The manuscript entitled “NLRP3 inflammasome is involved in cocaine-mediated potentiation on behavioral changes in CX3CR1 deficient mice” By Guo et al., described really interesting studies on the role of innate immune states on some cocaine addiction induced symptoms and markers of inflammation. All together the study presents a huge range of inflammation marker analysis which is very complete, and of very good interest for the field, but does address as much as they could the effect of either cocaine or the CX3CR1 gene deletion. Some consequences to other brain cells known to be very sensitive to inflammation could be put in light/perspectives.
Response: Thanks for the positive comments on our manuscript. Yes, we did observe astrocyte activation in CX3CR1 deficient mice (data not shown in this manuscript) and we added some discussion on astrocyte in the discussion section.
Concerns:
- Introduction: the direct relationship with ELS is perhaps premature, and could be addressed in the discussion section but not here. However further precisions on the innate immune system could be address in order for the reader to better understand the choices for biochemical analysis (for example gene expressions, microglial phenotypes, apoptosis).
Response: Yes. We removed the ELS description from Introduction part and added more descriptions on the innate immune system and NLRP3 inflammasome as suggested.
- Material and methods: the amount of time between the last-injection and/or the last behavioral experiment and the euthanasia is never addressed and should be. Justify why the cocaine dose and the number of injections is different with locomotor analysis and CPP tests. Therefore, we do not know what means “animals under cocaine” in each biochemical analysis, (how many injections? Dose? Days or hours between the last injection and the measurement?)
Response: Yes. Now we put more detailed description on cocaine injections and behavioral experiments in Material and method section. We employed 7-day cocaine injection regimen (20 mg/kg, i.p.) and 4-day unbiased CPP tests (10 mg/kg, i.p.) to explore the effects of CX3CR1 deficiency on cocaine-mediated behavioral changes. All biochemical analysis was performed by using the brain samples from mice with 7-day cocaine injections. The mice were sacrificed 24 hours after the last cocaine injection. The justifications on locomotor analysis and CPP were also added.
- CPP experiments are measuring “drug-seeking behavior” instead of “wanting effects” page 5 line 189. If there is experiment that includes some saline-treated animals, that is not the case for CPP experiments, why? Some controls are needed. In the same manner, there is no “cocaine result in”… you could not say something like that : there is no saline controls in any of the biochemical data in graphs of (figure 1, 2, 3)… The only figure 4 is presented without graphical representations to bring the assurance that there is no effects on some inflammation markers between WT and CX3CR1-/-. This is not convincing. Perhaps a two factor ANOVA (one factor treatment, one factor genotype), could be better for each structure, as the striatum and the hippocampus can be separated in the analysis and graphical representations. We need to know exactly which are the cocaine effects and which are only due to genotype and which are both…
Response: Sorry for the overlook! We changed the description “seeking effects” in CPP experiments. We had control mice with saline injections for CPP and now added them in the description. We did not see the difference between CX3CR1 and WT mice with saline injections (data were not shown). We understand the reviewer’s point: it is better to put all four groups (CX3CR1 deficiency and WT mice with cocaine or saline injections) together and analyze the results together using two factor ANOVA to obtain more convincing results. This is a really excellent suggestion. However, we found it was very difficult for us to run all samples in one SDS-PAGE gel to show the representative WBs results (16 samples, 4 groups X 4). We are extremely shortage on working hand due to the COVID-19 pandemic right now. After carefully thinking, we preferred to show the data and results in the current way. For other results, we did show all four groups data together in locomotor activity (Fig. 1A) and qRT-PCR experiments (supplementary Fig. 1).
- Figure 2: missing (E) in the legend, and there is two (C) and the graph in (C) is not in the legend also. Figure 5: I do not understand the (D) point.
Response: Sorry for the overlook. All errors have been corrected. We make it clearer on Figure 5 (D).
- Results/Discussion: The interleukines mRNA expression are evaluated in the brain, so we do not know for sure which is the kind of cell that increased its IL1 expression, be careful in your conclusions. Is there any difference or effect in living neurons or oligodendrocytes in the hippocampus and striatum following this huge inflammation after cocaine administration and or CX3CR1 deletion?
Response: Agree, other types of cell in the brain such as astrocytes also express IL1β and may contribute to the increased levels of IL1β in CX3CR1 -/- mice. Currently we do not know whether there is any difference or effect on neurons and/or oligodendrites in the striatum and in hippocampus following cocaine administration (we assumed they are) and CX3CR1 deficiency. This could be a new direction for our future studies.
Round 2
Reviewer 1 Report
The manuscript prepared by Guo et al. described the behavioral and biochemical data in WT and CX3CR1-/- mice with or without cocaine treatment. In their revision, some of the concerns raised by the reviewer had been addressed. However, there are still issues needed to be resolved.
Major issues
- The findings in CX3CR1-/- mice have to be described in detail before comparing the results between cocaine-treated WT and cocaine-treated CX3CR1-/- mice. In this case, the effects of CX3CR1 deficiency (first hit) would be identified and the additive effects of CX3CR1 deficiency and repetitive cocaine treatments (first + second hits) would be highlighted. A table that enlists the findings of each condition will be helpful!
- Repetitive cocaine treatments-produced enhanced locomotor activity is often referred to as cocaine-induced behavioral sensitization. The authors have to provide the actual data of locomotor activity after the first cocaine treatment.
- Before the CPP test, mice were examined in the three-chamber apparatus. The authors described, in the Methods and Materials section, that “mice with no pre-existing bias for either side of the chamber were selected for the next step (time spent in either chamber <80% of total time)”. Based on the description of calculating CPP scores, it is easy to estimate, in Figure 1B, the time spent of mice on each side. On average, WT mice spent about 62.5% of the time in the cocaine-paired chamber while CX3CR1-/- mice spent about 75% of the time in the cocaine-paired chamber!! In this regard, both WT and CR3CR1-/- showed no bias (<80% of total time) toward the cocaine-paired chamber.
- The concept of “microglial priming” is good and well accepted. However, the authors have to emphasize the fact or solid evidence in this and other studies. Besides, the authors have to use proper wording to describe this condition. For example, microglia prime? Non-prime microglia?
- The authors intended to use CX3CR1 deficiency, a non-naturally occurred condition, to simulate early life stress, a general idea that covers various biological/psychological insults and to link with drug addiction, a broad scenario in which various substances and symptoms are involved. The authors have to justify the value of CX3CR1 deficiency in the mechanistic discussion.
- English editing is necessary.
Minor issues
- CX3CR1 deficiency is a condition, usually, it is not used as a subject to start a sentence.
- Similarly, in lines 219-220, “the regulation of CX3CR1 deficiency on cocaine-mediated seeking effect” also needs to be rephrased.
- In the method of total RNA extraction, how to measure and take 100 mg brain tissue and directly add to 1 mL Trizol?
- In the text and figure legends, it is described as comparing the results obtained from “CR3CR1-/- mice” with “WT mice with cocaine administration (or at the basal condition)”. Please rephrase these statements.
- Citations are usually not listed in the abstract.
- Legend of figure 2, C and F, statistical results of WBs in A and E should be involved.
- Proofreading is necessary.
Author Response
Major issues
- The findings in CX3CR1-/- mice have to be described in detail before comparing the results between cocaine-treated WT and cocaine-treated CX3CR1-/- mice. In this case, the effects of CX3CR1 deficiency (first hit) would be identified and the additive effects of CX3CR1 deficiency and repetitive cocaine treatments (first + second hits) would be highlighted. A table that enlists the findings of each condition will be helpful!
Response: Good point! A table has been made and added as supplementary table 1.
- Repetitive cocaine treatments-produced enhanced locomotor activity is often referred to as cocaine-induced behavioral sensitization. The authors have to provide the actual data of locomotor activity after the first cocaine treatment.
Response: Yes. Now we included the locomotor data for the first cocaine injection as supplementary Fig. 7.
- Before the CPP test, mice were examined in the three-chamber apparatus. The authors described, in the Methods and Materials section, that “mice with no pre-existing bias for either side of the chamber were selected for the next step (time spent in either chamber <80% of total time)”. Based on the description of calculating CPP scores, it is easy to estimate, in Figure 1B, the time spent of mice on each side. On average, WT mice spent about 62.5% of the time in the cocaine-paired chamber while CX3CR1-/- mice spent about 75% of the time in the cocaine-paired chamber!! In this regard, both WT and CR3CR1-/- showed no bias (<80% of total time) toward the cocaine-paired chamber.
Response: For CPP, we firstly calculated the scores (the time difference between saline-paired environment and cocaine-paired environment) of each mouse and then compared whether the scores from WT and CX3CR1 -/- mice have significant difference. We did not compare the average percentage of time in cocaine-paired environment between groups. WT and CR3CR1-/- may not show the bias (<80% of total time) toward the cocaine-paired chamber but they did show the significant difference on the time spent in cocaine-paired chamber.
- The concept of “microglial priming” is good and well accepted. However, the authors have to emphasize the fact or solid evidence in this and other studies. Besides, the authors have to use proper wording to describe this condition. For example, microglia prime? Non-prime microglia?
Response: Yes, we included references as solid evidence to address the critical roles of microglia priming in other studies (reference 45, 46). We used “microglia prime” in our manuscript.
- The authors intended to use CX3CR1 deficiency, a non-naturally occurred condition, to simulate early life stress, a general idea that covers various biological/psychological insults and to link with drug addiction, a broad scenario in which various substances and symptoms are involved. The authors have to justify the value of CX3CR1 deficiency in the mechanistic discussion.
Response: Excellent point. We justified the relevance of CX3CR1 deficiency for drug addiction study in the discussion section.
- English editing is necessary.
Response: Yes. We requested the English editing.
Minor issues
- CX3CR1 deficiency is a condition, usually, it is not used as a subject to start a sentence.
Response: Thanks. We requested English editing for improving this.
- Similarly, in lines 219-220, “the regulation of CX3CR1 deficiency on cocaine-mediated seeking effect” also needs to be rephrased.
Response: Yes. We requested English editing for improving this.
- In the method of total RNA extraction, how to measure and take 100 mg brain tissue and directly add to 1 mL Trizol?
Response: Actually, it was a range 50 - 100 mg. We changed that in the context.
- In the text and figure legends, it is described as comparing the results obtained from “CR3CR1-/- mice” with “WT mice with cocaine administration (or at the basal condition)”. Please rephrase these statements.
Response: Yes. We requested English editing for improving this.
- Citations are usually not listed in the abstract.
Response: Yes. We removed the citation in the abstract.
- Legend of figure 2, C and F, statistical results of WBs in A and E should be involved.
Response: Sorry for the confusion. We should make it clear on the legend. Actually, the statistical results of C were from Fig. A and Fig. B and statistical results of F were from Fig. D and Fig. E. Now, they were corrected.
- Proofreading is necessary.
Response: We request the English editing for our manuscript.